# Osteocyte-Related Cytokines Regulate Osteoclast Formation and Bone Resorption

**DOI:** 10.3390/ijms21145169

**Published:** 2020-07-21

**Authors:** Hideki Kitaura, Aseel Marahleh, Fumitoshi Ohori, Takahiro Noguchi, Wei-Ren Shen, Jiawei Qi, Yasuhiko Nara, Adya Pramusita, Ria Kinjo, Itaru Mizoguchi

**Affiliations:** Division of Orthodontics and Dentofacial Orthopedics, Tohoku University Graduate School of Dentistry, 4-1, Seiryo-machi, Aoba-ku, Sendai, Miyagi 980-8575, Japan; marahleh.aseel.mahmoud.t6@dc.tohoku.ac.jp (A.M.); fumitoshi.ohori.t3@dc.tohoku.ac.jp (F.O.); takahiro.noguchi.r4@dc.tohoku.ac.jp (T.N.); shen.wei.ren.t5@dc.tohoku.ac.jp (W.-R.S.); qi.jiawei.p8@dc.tohoku.ac.jp (J.Q.); yasuhiko.nara.q6@dc.tohoku.ac.jp (Y.N.); adya.pramusita.q6@dc.tohoku.ac.jp (A.P.); ria.kinjou.p5@dc.tohoku.ac.jp (R.K.); mizo@tohoku.ac.jp (I.M.)

**Keywords:** osteoclast, osteocyte, cytokine, bone

## Abstract

The process of bone remodeling is the result of the regulated balance between bone cell populations, namely bone-forming osteoblasts, bone-resorbing osteoclasts, and the osteocyte, the mechanosensory cell type. Osteoclasts derived from the hematopoietic stem cell lineage are the principal cells involved in bone resorption. In osteolytic diseases such as rheumatoid arthritis, periodontitis, and osteoporosis, the balance is lost and changes in favor of bone resorption. Therefore, it is vital to elucidate the mechanisms of osteoclast formation and bone resorption. It has been reported that osteocytes express Receptor activator of nuclear factor κΒ ligand (RANKL), an essential factor for osteoclast formation. RANKL secreted by osteocytes is the most important factor for physiologically supported osteoclast formation in the developing skeleton and in pathological bone resorption such as experimental periodontal bone loss. TNF-α directly enhances RANKL expression in osteocytes and promotes osteoclast formation. Moreover, TNF-α enhances sclerostin expression in osteocytes, which also increases osteoclast formation. These findings suggest that osteocyte-related cytokines act directly to enhance osteoclast formation and bone resorption. In this review, we outline the most recent knowledge concerning bone resorption-related cytokines and discuss the osteocyte as the master regulator of bone resorption and effector in osteoclast formation.

## 1. Introduction

Bone is a dynamically changing tissue that is continuously degraded and built via the process of bone remodeling, the process in which bone cell populations achieve a balance between resorption and deposition episodes [1]. The process of bone remodeling is the result of the regulated balance between bone-forming osteoblasts, bone-resorbing osteoclasts, and the osteocyte, the mechanosensory cell type [2].

Osteoclasts derived from the hematopoietic stem cell lineage are the principal cells responsible for bone resorption [3]. The balance in osteolytic diseases, such as rheumatoid arthritis, periodontitis, and osteoporosis is lost, favoring bone resorption [4]. Molecular signals act together with cellular components to regulate bone resorption. Macrophage colony-stimulating factor (M-CSF) is the first essential cytokine that induces osteoclast formation by binding to the c-fms receptor and promotes osteoclast precursor differentiation and maturation [5]. Receptor activator of nuclear factor-κB ligand (RANKL) is a member of the tumor necrosis factor superfamily, which is secreted by osteoblasts, bone marrow stromal cells [6], and lymphocytes [7]. RANKL interacts with the RANK receptor on the surface of osteoclast precursors and promotes their differentiation into bone-resorbing osteoclasts. 

Osteocytes get embedded in the osteoblast-secreted matrix during their differentiation from an osteoblast to osteocyte [8]. Osteocytes reside within lacunae and they comprise 90% of the bone cell population. Osteocytes communicate with each other and other cell types through dendrites extending to the bone surface [9]. The human body has about 42 billion osteocytes with an average half-life of 25 years [10].

Osteocytes have been shown to function as regulators of mineral metabolism and perilacunar matrix remodeling and as mechanosensory cells [11]. It has been established that osteocytes express RANKL, and osteocyte-secreted RANKL is the most important for physiologically supported osteoclast formation in the developing skeleton [12,13]. An osteoporotic phenotype becomes increasingly apparent in osteocyte-specific RANKL-deficient mice postnatally. This result suggests that RANKL secreted by osteocytes plays a critical role postnatally. Furthermore, osteocytes induce osteoclast formation in cocultures with bone marrow cells upon addition of prostaglandin E2 (PGE2) and vitamin D (1,25(OH)_2_D_3_) [12]. Osteocyte-specific RANKL-deficient mice subjected to unloading of mechanical force do not experience bone loss, as observed in wildtype mice, and are protected against such bone loss [13]. It has also been reported that experimental periodontal bone loss is attenuated in mice with osteocyte-deficient RANKL [14]. Deletion of osteocyte RANKL confers an increase in cancellous bone mass in osteogenesis imperfecta mice [15]. These results suggest that osteocyte RANKL affects bone resorption in both healthy and diseased individuals. In our recent study, we found that tumor necrosis factor-α (TNF-α) directly enhances osteocyte RANKL expression and promotes osteoclast formation [16] and that mechanical force induces TNF-α, which leads to sclerostin expression in osteocytes and sclerostin subsequently enhances osteoclast formation [17]. These results suggest that osteocyte-related cytokines affect osteoclast formation and bone resorption.

Apart from the classic pathway of bone resorption which utilizes the action of osteoclasts, osteocytes have been described as capable of bone resorption and expanding their lacuno-canalicular housing. This concept has been termed osteocytic osteolysis, which has been observed in a number of settings: in lactating mice [18] and high calcium demands [19], through sclerostin signaling [20], through calciotropic hormones signaling such as 1,25(OH)_2_D_3_ [21] and parathyroid hormone [22], and in the absence of physiologic loading [23]. Osteocytic osteolysis could be of importance for the maintenance of the osteocyte perilacunar space, adaptation to different loading situations, and it might contribute to global calcium homeostasis; however, the importance of this process is still enigmatic, and its relative importance to direct osteoclastic bone resorption is a subject of study.

Osteocytes control bone formation through sclerostin regulation, which inhibits the Wnt/β-catenin pathway, a major osteoblastogensis mediator. Loss of sclerostin causes a high bone mass phenotype such as in sclerosteosis and Van Buchem’s disease [24]. Sclerostin is a crucial link between osteoblasts and the mechanosensory ability of osteocytes, as absence of loading leads to increased sclerostin expression and bone loss, which is evident in mice that do not gain bone mass with loading due to their inability to downregulate sclerostin [25]. Osteocytes also control osteogenesis through the fine tuning of their oxygen sensing apparatus prolyl hydroxylase 2 (PHD 2), in which silencing of PHD 2 causes an increase in bone mass and strength and protects against bone loss caused by estrogen deficiency and absence of mechanical loading, through decreased expression of sclerostin [26]. Parathyroid hormone is also a regulator of sclerostin expression in osteocytes and acts as an anabolic agent when administered intermittently [27]. Anti-sclerostin antibody may be a promising bone anabolic agent, and further studies should elucidate the mechanism by which precise targeting of sclerostin could mediate bone anabolism.

In this review, we outline the recent knowledge concerning bone resorption-related cytokines and discuss the osteocyte as the master regulator of bone resorption and effector in osteoclast formation. We further discuss the possible mechanisms by which osteocyte-related cytokines regulate bone resorption.

## 2. Osteoclast Regulatory Cytokines

### 2.1. Osteoclastogenic Cytokines

Osteoclasts are responsible for bone resorption in physiological bone remodeling and pathological bone destruction that accompanies osteolytic inflammatory diseases. Inflammation is a hallmark of various pathological conditions such as those confined locally to the bone, including rheumatoid arthritis [28] and periodontitis [29]. Moreover, systemic inflammation is involved in a myriad of conditions such as kidney disease [30], inflammatory bowel disease [31], chronic skin inflammation [32], bacterial infection [33], cancer [34], acute injury [35], metabolic syndrome in obese patients [36], and diabetes [37]. Proinflammatory cytokines secreted in the bone vicinity enter blood circulation easily and travel to distant organs, causing systemic effects. Many inflammatory cytokines have osteoclastogenic effects on bone. In vivo immune and inflammatory responses are regulated by a complex network of cytokines. In rheumatoid arthritis, TNF-α, Interleukin (IL)-1, IL-6, and IL-17 produced by synovial macrophages and T cells act on osteoblasts to promote RANKL expression [38]. There are many reports on osteoclastogenesis-promoting cytokines including IL-1 [39], IL-6 [40], IL-7 [41], IL-8 [42], IL-11 [43], IL-15 [44], IL-17 [45], IL-23 [46], IL-34 [47], and transforming growth factor-β [48].

### 2.2. Anti-Osteoclastogenic Cytokines

Numerous cytokines have anti-osteoclastogenic and anti-resorptive effects on bone. IL-3 inhibits osteoclast formation and bone resorption via inhibition of c-fos [49]. IL-4 attenuates RANKL-induced osteoclast formation by inhibiting NFATc1 via NF-κB inactivation activation [50,51] and TNF-α-induced osteoclast formation in vitro and in vivo [52,53]. IL-10 inhibits osteoclast formation directly by inhibition of NFATc1 [54]. IL-12 [55,56,57,58] and IL-18 [59,60,61] also inhibit osteoclast formation and bone resorption. They act in synergy to induce apoptosis of osteoclast precursors and inhibit TNF-α-mediated osteoclastogenesis in myeloid cells. TNF-α induces Fas expression, while IL-12 and IL-18 induce FasL expression, leading to apoptosis of osteoclast precursors [57,58,60,61]. IL-12 also inhibits lipopolysaccharide (LPS)-induced osteoclast formation in vivo and induces apoptosis of osteoclasts [62]. IL-13 [63] and IL-27 inhibit RANKL-induced osteoclast formation by STAT1-dependent inhibition of c-Fos [64]. IL-33 attenuates RANKL-induced osteoclast formation by modulation of BLIMP1 and interferon regulatory factor 8 expression and inhibition of IκB phosphorylation and NF-κB nuclear translocation [65], as well as mechanical loading-induced osteoclast formation [66]. Interferon (IFN)-α [67], IFN-β [68], and IFN-γ [48,69] also inhibit osteoclastogenesis. IFN-γ inhibits LPS-induced osteoclast formation by inhibition of RANK and c-fos in osteoclast precursors via the TLR signaling pathway [70]. Furthermore, when IFN-γ is introduced into TNF-α-induced osteoclast formation from bone marrow cells, TNF-α induces Fas expression, and IFN-γ induces FasL expression to induce apoptosis of osteoclast precursors [71].

## 3. Osteocyte-Related Cytokines

The osteocyte secretes a diverse array of cytokines and signaling molecules including sclerostin, RANKL, osteoprotegerin (OPG), TNF-α, IL-1β, IL-6, fibroblast growth factor (FGF) 23, and insulin-like growth factor (IGF)-1 [11,28,72]. These molecules exert their effects on the osteocyte itself, other bone cells and cells of the immune system harboring in bone, and distant organs, functioning in autocrine, paracrine, and endocrine fashions, respectively. Inflammation both locally and systemically has been reported to affect osteocyte survival and activity. Apoptotic osteocytes signal to nearby osteocytes and macrophages to secrete proinflammatory molecules and growth factors such as RANKL, TNF-α, IL-1β, IL-6, IL-8, and vascular endothelial growth factor (VEGF) [28,73]. Osteocytes respond to physiological levels of proinflammatory cytokines, maintaining their survival and functions. However, inflation of inflammatory secretion regulates osteocyte-derived molecules and cytokines in a way that contributes to bone and non-bone-related pathologies. Osteocytes respond to single or multiple cytokines acting synergistically. Osteocytes secrete inflammatory cytokines that regulate osteocyte functions and other bone cells, and amplify inflammation at a distance [28]. This property grants the osteocyte the status of a true endocrine cell type.

### 3.1. RANKL and OPG

RANKL-RANK signaling activates osteoclast differentiation and functions, and inhibits osteoclast apoptosis. As a result, this dynamic induces bone resorption. However, OPG, which is a RANKL decoy receptor, prevents RANKL-RANK binding [74]. Osteocytes are a significant source of RANKL that induces osteoclastogenesis and bone resorption [12,13]. Osteocytes also express OPG that is downregulated significantly in osteocyte-specific β-catenin-deficient mice. Osteocyte OPG-deficient mice show a low bone mass phenotype, suggesting that osteocyte specific OPG plays an important role in bone mass regulation [75]. Osteocytes participate in the development of periodontitis and periodontitis-induced alveolar bone loss by upregulating proinflammatory cytokine secretion. Osteocyte-derived RANKL is elevated in diabetic rats with periodontitis, which correlates well with a high osteoclast number, osteoclast activity, and bone resorption [14,76]. Another osteocyte-secreted molecule, sclerostin, increases the RANKL/OPG ratio in periodontitis via ERK1/2-MAPK in alveolar bone [77]. Osteocyte-RANKL expression is also governed by Cx43, a protein that allows the formation of gap junctions between osteocytes forming a connected network of cells. Cx43 expression in osteocytes decreases with age [78]. In mice lacking Cx43, osteocyte apoptosis increases [79]. These results suggest that channeling through Cx43 controls osteocyte survival [79]. Decreased Cx43 expression reduces prosurvival microRNA-21 and promotes osteocyte apoptosis. Therefore, deletion of Cx43 increases osteocyte RANKL and osteoclast formation [80]. Multiple myeloma, a plasma cell malignancy, is characterized by increased bone resorption. Multiple myeloma cells induce osteocyte apoptosis that increases osteocyte-derived sclerostin and the RANKL/OPG ratio, resulting in osteoclast formation and bone resorption [81]. It has been reported that zoledronic acid and plumbagin-loaded nanoparticles target osteocytes to release plumbagin, decreasing osteocyte RANKL expression. The particles attenuate tumorigenesis and osteoclast formation in a breast cancer bone metastasis model by inhibiting RANKL and sclerostin expression in osteocytes [82]. One of the common comorbidities in inflammatory bowel disease is bone loss, leading to an elevated fracture risk [83]. Inflammatory bowel disease induced by an enema of 2,4,6-trinitrobenzenesulfonic acid in mice induces systemic inflammation and alters secretion of proteins from osteocytes, such as RANKL and OPG, increasing the RANKL/OPG ratio and bone resorption [31]. Spinal cord injury patients have 40–70% lower bone mass in cancellous-rich bone sites and 25–35% lower bone mass in cortical sites such as the shafts of the tibia [84]. Rodent models of spinal cord injury show similar bone loss [85]. Spinal cord injury in a rat model induces RANKL and OPG expression in osteocytes, altering the RANKL/OPG ratio in osteocytes [86]. Microcracks of the bone in knee injury remodel via RANKL expressed in osteocytes [87]. Serum RANKL also increased in rodent thermal injury model compared to that of sham burn with bone mineral density as well as weight bearing capacity being negatively influenced acutely and several weeks after the injury [88]. Optimal mechanical loading induces osteocyte functions and survival. However, excessive force on bone induces osteocyte apoptosis and expression of RANKL in osteocytes, which induces osteoclast formation and bone resorption [89]. It is apparent that osteocyte RANKL expression is induced via local and systemic routes, which in turn also functions locally, affecting nearby cells or systemically in models of systemic inflammatory diseases. RANKL is expressed as a membrane-bound protein (mRANKL) that can be cleaved and shed into the circulation as soluble RANKL (sRANKL); elevated sRANKL has been linked to local [90] and systemic inflammatory conditions [91,92] as well as cancer cohorts [93]. Both mRANKL and sRANKL are important for osteoclast formation. This is evident in mice lacking sRANKL, in which osteoclast number was reduced and cancellous bone volume increased with increased age, but it was indistinguishable from that of wildtype mice at a younger age; despite sRANKL’s contribution to physiologic bone resorption, sRANKL had no protective effect on reduced bone mass in ovariectomized sRANKL-lacking mice, as those mice experienced similar bone loss to wildtype ovariectomized mice [94]. Cultures of osteoclast precursor cells in an osteocyte conditioned medium failed to produce osteoclasts, highlighting the importance of mRANKL for osteoclastogenesis in culture conditions [12]. Conversely, conflicting evidence of coculture of bone marrow cells with a conditioned medium of mechanically damaged MLO-Y4 that produce sRANKL and soluble M-CSF were able to mediate osteoclastogenesis [95]. While the role of mRANKL is clear and documented, further studies are needed to clarify the mechanism and significance of RANKL shedding in osteocytes.

### 3.2. TNF-α

TNF-α is a potent proinflammatory cytokine that is secreted by various cell types, such as mononuclear phagocytes, in which it activates cytocidal functions, playing a major role in host defense. TNF-α is central to the pathogenesis of disorders involving inflammation [96], such as obesity, which is characterized by chronic inflammation possibly via upregulation of TNF-α that activates the inflammatory cascade and affects various organ functions [36,97]. Furthermore, obesity is linked to decreased bone formation and increased bone resorption through TNF-α-induced upregulation of osteocyte-derived sclerostin and RANKL [98]. TNF-α upregulates sclerostin expression in the MLO-Y4 cell line, a murine osteocyte-like cell line. This effect is inhibited by suppression of NF-κB signaling using NF-κB p65 siRNA [98]. These results suggest that TNF-α upregulates sclerostin expression via NF-κB signaling activation, contributing to bone loss. It is well known that TNF-α is expressed during periodontitis [99]. Patients suffering from obesity in addition to periodontitis have a higher level of TNF-α expression than patients with periodontitis only [100]. Infliximab, which is a TNF-α antagonist, inhibits sclerostin and RANKL expression in osteocytes and attenuates alveolar bone loss in periodontitis rats with diabetes [76]. Under chronic hyperglycemic conditions, mRNA and protein levels of sclerostin, reactive oxygen species, and TNF-α expression levels increase in MLO-Y4 cells. N-Acetylcysteine treatment or knockdown of TNF-α inhibits this high glucose-induced sclerostin expression. These results suggest that hyperglycemia increases sclerostin expression by inducing reactive oxygen species and TNF-α expression and that regulation of TNF-α and oxidative stress levels may be a valuable therapeutic strategy for alveolar bone complications in chronically hyperglycemic patients [101]. High mobility group box 1 protein released from damaged MLO-Y4 cells induces RANKL, TNF-α, and IL-6 from other stromal cells and macrophages [73,102]. These results suggest that osteoclast formation and bone resorption are induced around apoptotic osteocyte regions. Rheumatoid arthritis is a chronic autoimmune disease with local joint inflammation characterized by joint space narrowing, local bone erosion, and extra-articular manifestation. Serum and synovial fluid of patients with rheumatoid arthritis have high amounts of proinflammatory cytokines such as TNF-α, IL-1β, and IL-6 [103]. Human osteocyte-enriched cells were cultured with serum of active rheumatoid arthritis patients ex vivo, which increased IL-1β, TNF-α, SOST, and DKK1 gene expression levels [28]. These results highlight the role of the osteocyte as a new therapeutic target to inhibit bone destruction in rheumatoid arthritis patients. In our recent study, we evaluated the direct effects of TNF-α on osteocytes. In the study, we used highly purified primary osteocytes that were isolated by cell sorting from neonatal calvariae of DMP1-Topaz mice expressing green fluorescent protein under the control of the dentin matrix protein 1 promoter [16]. Primary osteocytes cultured with TNF-α showed significantly higher RANKL mRNA expression. Additional in vivo experiments of TNF-α injected into the mouse calvaria showed increases in the osteoclast number and RANKL-positive osteocytes. Furthermore, osteocytes cultured with TNF-α showed upregulation of ERK1/2, P38, and JNK MAPK phosphorylation measured by western blotting. Inhibition of ERK1/2, p38, and JNK MAPKs via their respective inhibitors (U0126, SB203580, and SP600125) in osteocytes cultured with TNF-α suppressed RANKL mRNA expression to baseline levels compared with untreated cells. We also found that TNF-α activates the NF-κB pathway in osteocytes measured by p65 subunit nuclear translocation [16]. Activated AKT also acts as an anti-apoptotic signal, and TNF-α stimulates the AKT pathway in a cell-type-specific manner [104]. We found a trend of activated AKT phosphorylation to levels that did not reach statistical significance [16]. The results of the study supported the role of osteocytes in their potential contribution to bone destruction by increasing RANKL expression via the direct action of TNF-α. Osteocytes may be a therapeutic target when inflammation is central to a disease pathology.

### 3.3. IL-1β

IL-1β is an important proinflammatory cytokine that is normally produced by dendritic cells, monocytes, T cells, and macrophages [105]. IL-1β enhances osteoclast formation and bone resorption in rheumatoid arthritis, periodontal diseases, and osteoporosis [106,107,108]. Treatment with exogenous cytokines, such as IL-1β and TNF-α, and in combination of IL-6, upregulates IL-1β expression in human osteocytes [28]. It has been reported that IL-1β enhances RANKL expression and inhibits OPG expression in MLO-Y4 cells. IL-1β-treated osteocytes increase the formation of osteoclasts. However, conditioned medium from mechanically loaded osteocytes via pulsatile fluid flow and treated with IL-1β prevents osteoclast formation. Thus, mechanical loading of osteocytes abolishes IL-1b-induced osteoclast formation [109]. In diabetic rats with periodontitis, bone destruction correlates with elevated osteocyte-derived IL-1β expression [14,76,110]. Lenalidomide is a drug that has shown potential in anti-cancer and anti-inflammatory treatments. Lenalidomide treatment rescues IL-1β-induced osteocyte apoptosis and inhibits IL-1β-induced RANKL and Sclerostin expression in MLO-Y4 osteocytes. Conditioned medium from lenalidomide-treated osteocytes inhibits osteoclast formation and bone resorption in vitro. Furthermore, IL-1β-induced IκB degradation is remarkably downregulated by lenalidomide in MLO-Y4 osteocytes. These results show that lenalidomide regulates NF-κB signaling in MLO-Y4 osteocytes. In vivo analysis of MLO-Y4 osteocyte apoptosis and osteoclast formation in an osteoarthritis mouse model showed that lenalidomide inhibits osteocyte apoptosis and RANKL-induced osteoclast formation [111]. These results reveal the importance of IL-1β/NF-κB signaling in osteocytes to attenuate osteoclast formation in vitro and in vivo.

### 3.4. IL-6

IL-6 is a pleotropic cytokine functioning as a proinflammatory cytokine and in regulation of the immune system. IL-6 has important roles in inflammation, autoimmunity, injury, hematopoiesis, diabetes, atherosclerosis, rheumatoid arthritis, and cancer [112,113,114]. It is synthesized by a wide variety of cell types including monocytes, macrophages, T cells, B cells, fibroblasts, keratinocytes, endothelial cells, mesangial cells, adipocytes, and some tumor cells [115,116]. IL-6 exerts an indirect effect on osteoclasts, promoting osteoclast activity and bone resorption by inducing RANKL expression in osteoblasts [117]. IL-6-overexpressing transgenic mice show osteopenia with increases in the osteoclast number and activity [118]. Furthermore, IL-6-neutralizing antibodies inhibit TNF-α and IL-1β-stimulated osteoclast formation [119]. Therefore, it is recognized that IL-6 is a promoter of osteoclastic bone resorption and centrally involved in the pathogenesis of bone loss in chronic and acute inflammation [118], periodontitis [120], rheumatoid arthritis, and osteoporosis [121]. It has been reported that treatment with exogenous proinflammatory cytokines, such as IL-1β and TNF-α, as well as combinations of IL-1β, TNF-α, and IL-6, upregulates IL-6 expression in a human osteocyte-rich cell fraction [28]. Brucellosis caused by *Brucella abortus* infection results in bone loss [122]. MLO-Y4 osteocytes infected with *B. abortus* upregulate expression of IL-6. The culture supernatants of these *B. abortus*-infected osteocytes induce osteoclast formation [33]. LPS is a bacterial toxin that causes inflammation and significantly increases the production of proinflammatory cytokines including IL-6 [123]. IL-6 expression in MLO-Y4 osteocytes is increased by LPS via activation of the ERK1/2 signaling pathway [123]. IL-6 enhances osteocyte-mediated osteoclast formation through upregulation of RANKL and JAK2 activities [124]. In diabetics and elderly patients, high glucose levels and advanced glycation end products are associated with disrupted functions of bone cells and deterioration of bone mass, inducing osteoporosis. Advanced glycation end products activate MAPKs ERK1/2 and P38 and STAT3 signaling through upregulation of osteocyte-derived IL-6 and VEGF-A secretion [125] as well as osteocyte apoptosis [125,126]. IL-6 is also one the cytokines secreted by apoptotic osteocytes, which signals near osteocytes and macrophages to release other proinflammatory cytokines and growth factors [73].

### 3.5. FGF23

FGF23 is considered as a hormone belonging to the FGF family of proteins. There are 22 FGF family members in humans, which are categorized into three subfamilies according to their mode of action, autocrine, paracrine, or endocrine. FGF23 belongs to a subfamily together with FGF19 and FGF21 that exert their functions in an endocrine manner [127]. FGF23 is produced in bone by osteoblasts, but mainly by osteocytes [128]. The principal documented actions of FGF23 in mineral homeostasis are the reduction of serum phosphate and 1,25(OH)_2_D_3_ as well as parathyroid hormone levels [129,130]. *Fgf23* was identified as the gene responsible for autosomal dominant hypophosphatemic rickets [131]. *Fgf23* knockout mice were established to study the physiological actions of FGF23. These mice show hyperphosphatemia with enhanced proximal tubular phosphate reabsorption and a high 1,25(OH)_2_D_3_ level [132]. An increase in the FGF23 level is recognized in the pathogenesis of secondary hyperparathyroidism with low 1,25(OH)_2_D_3_, hyperphosphatemia, and hypocalcemia in patients with advanced chronic kidney diseases [133]. Osteocyte production of FGF23 is the main route for mineral and phosphate homeostasis. Other highly expressed osteocytic genes, such as PHEX [134], Dmp1 [135], and MEPE [136], participate in the regulation of mineral and phosphate homeostasis either directly or by regulating FGF23 signaling. Loss of the functions of either Dmp1 or PHEX dramatically increases FGF23 production that increases phosphate excretion, resulting in osteomalacia and rickets [135]. However, the mechanism by which PHEX regulates FGF23 levels is not fully understood since FGF23 is not a direct substrate for PHEX, suggesting that another substrate or other indirect downstream pathways link PHEX with FGF23 levels. PHEX has been shown to alter the expression but not the degradation of FGF23 [137]. MEPE-null mice have increased bone mass due to loss of the action of the acidic serine aspartate-rich MEPE-associated motif (ASARM), a potent inhibitor of mineralization [136,138]. MEPE binds to PHEX, which prevents the release of ASARM and prevents the downregulation of FGF23. If ASARM is released, it binds to PHEX and prevents the enzymatic activity of PHEX leading to upregulation of FGF23, which may also provide another mechanism by which PHEX controls FGF23 levels [138,139]. The direct effect of FGF23 on osteoclasts has been described as biphasic. FGF23 inhibits osteoclast differentiation in the early stages of in vitro culture of monocytes together with RANKL and M-CSF. However, this effect diminishes when FGF23 is added at a later stage of culture and when monocytes are treated with a pan-FGF receptor inhibitor. In contrast, FGF23 increases osteoclast activity as measured by the degree of resorption area per well or per osteoclast. This effect is only evident with low doses of FGF23 in vitro [140]. The effect of FGF23 on osteoclasts is reported to be independent of klotho, which is an FGF23 and FGFRc1 binding protein, but this does not rule out that klotho might be involved in FGF23 effects on osteoclasts, because klotho is both a transmembrane protein and soluble protein in circulation [140]. Other studies have reported that FGF23 has no effect on osteoclast formation in wildtype bone marrow cultures [141] and reasoned that the decrease in osteoclast formation in FGF23-deficient mice is due to deficient parathyroid hormone actions [132]. In transgenic mice overexpressing FGF23, the number of osteoclasts, serum level of TRACP 5b, and mRNA levels of TRAP and cathepsin K are unchanged, but markers of bone matrix degradation are elevated. However, FGF23 transgenic mice exhibit structural and morphological changes in osteoclasts with an immature ruffled border and clear zone, despite the fact that the levels of MMP-9 and cathepsin k around the morphologically aberrant osteoclasts are comparable with those in wildtype osteoclasts, indicating that they are functionally sound in terms of resorptive activity [142]. FGF23 is considered as a marker of numerous conditions such as chronic kidney disease, in which osteocyte sclerostin and FGF23 are elevated [143]. Conditions in which serum phosphate alone or in combination with 1,25(OH)_2_D_3_ are elevated also exhibit an increase in FGF23 mRNA expression by murine osteocytes [144] as well as proinflammatory mediators TNF-α, IL-1β, and LPS that also increase FGF23 expression by osteocytes [145]. TNF-α and IL-1β-mediated upregulation of FGF23 in osteocytes is dependent on activation of the NF-κB pathway [145]. TNF-α and IL-8 treatment enhances FGF23 gene expression in human osteocyte cultures [28]. A combination of IL-1β, IL-6, and TNF-α treatments synergistically upregulates FGF23 gene expression [28]. A correlation between elevated levels of serum FGF23 in rheumatoid arthritis (RA) patients with disease activity and bone resorption has been established [146] because serum RA enhances osteocyte-mediated osteoclastogenesis [28].

### 3.6. IGF-1

IGF-1 is polypeptide hormone that is primarily produced by liver cells (75%) following stimulation by growth hormone (GH) [147]. Except for the liver, almost every organ expresses IGF-1 receptors under physiological conditions. The GH/IGF-1 axis is mainly responsible for increasing muscle mass, lipolysis, and bone growth [148]. It has been shown that osteocytes produce a considerable amount of IGF-1 that regulates osteoblasts and osteoclasts in paracrine or autocrine manners [149]. Deletion of the *Igf1* gene in osteocytes causes an impairment in developmental bone growth [150]. Mechanical loading induces local IGF-1 expression of osteocytes [151], and then osteocyte-derived IGF-1 immediately induces osteogenic differentiation by upregulating Wnt10b expression and suppressing SOST expression in an autocrine manner [152]. This indicates a role of osteocyte-derived IGF-I in the translation of mechanical stimuli into bone formation. IGF-1 enhances osteoblast proliferation via MAPK and Akt pathways [153]. Osteogenic differentiation of periodontal ligament stem cells can be induced by IGF-1 treatment via activation of ERK and JNK MAPK pathways [154]. However, the anabolic bone effects of parathyroid hormone are synergized by IGF-1 in vivo [155]. Osteoblast-to-osteocyte transition is also enhanced by IGF-1-induced parathyroid hormone receptor phosphorylation [156]. Moreover, local IGF-1 derived from osteocytes is important for osteoblast-osteoclast interactions and osteoclast formation [157]. IGF-1 promotes osteoclast differentiation by increasing both M-CSF and RANKL expression in osteoblasts and RANK expression in osteoclasts [158]. In addition, IGF-1-induced osteoblast and osteoclast differentiation is regulated by ephrin B2/EphB4-mediated cell–cell communication [159].

### 3.7. IL-8

IL-8 is a CXC chemokine known to play an important role in regulation of the inflammatory response. It was identified as a neutrophil chemotactic factor. IL-8 is produced by macrophages, epithelial cells, airway smooth muscle cells, and endothelial cells. IL-8 exerts its effects by binding to its receptors, CXCR1 and CXCR2 [160]. It is overexpressed in many tumors and cancer cell lines, and promotes tumor growth, angiogenesis, and metastasis in a variety of human cancers [161]. It is also elevated in the serum and synovial fluid of RA patients [162]. Production of IL-8 by breast cancer cells increases osteoclast formation and may contribute to bone metastasis [163]. IL-8 is a potential stimulator of osteoclast differentiation and bone resorption. IL-8 upregulates RANKL expression in osteoblasts and induces osteoclastogenesis directly by binding to CXCR1 on cells [42]. It has also been reported that IL-8 plays an important role as a RANKL-induced autocrine mediator of osteoclast formation. A neutralizing anti-IL-8 antibody inhibits osteoclast formation and NFATc1 nuclear translocation [164]. IL-8 is a direct factor that induces osteoclast differentiation in multiple myeloma [165]. Bone marrow stromal cells from patients with multiple myeloma enhance production of IL-8. IL-8 potentiates NF-κB activation induced by certain multiple myeloma bone marrow stromal cells. Therefore, IL-8 production in myeloma promotes tumor growth and contributes to increased bone resorption [166]. Binding of anti-citrullinated protein antibodies to osteoclasts induces production of IL-8 and enhances osteoclast maturation and activation [167]. IL-8 also stimulates RANKL expression in bone marrow stromal cells. These results suggest that IL-8 not only induces osteoclast differentiation directly, but also RANKL expression contributing to osteoblast-mediated osteoclastogenesis [168]. Bone-metastasized cells from primary cancers such as lung and breast cancers release cytokines, such as IL-6, IL-8, M-CSF, and monocyte chemotactic protein-1, which affect osteocyte functions [169]. IL-8 treatment enhances TNF-α, IL-8, and FGF23 gene expression and TNF-α reciprocates by enhancing IL-8 gene expression in osteocyte-enriched cells. IL-8 gene expression is also upregulated by the synergistic action of TNF-α, IL-1β, and IL-6 [28].

### 3.8. M-CSF

M-CSF was identified as a hematopoietic cell growth factor that induces differentiation of macrophages from bone marrow progenitors. M-CSF as a small protein is produced constitutively by various cell types such as macrophages, endothelial cells, fibroblasts, osteoblasts, lymphocytes, monocytes, and osteocytes [170]. M-CSF induces proliferation, differentiation, and survival of monocytes, macrophages, and bone marrow progenitor cells. M-CSF is indispensable for the proliferation and differentiation of osteoclast precursors [171]. In osteopetrotic op/op mice, a thymidine insertion in the *Csf-1* results in M-CSF deficiency, which are deficient for osteoclasts and macrophages. This deficiency is caused by the absence of functional M-CSF and rescued by injection of M-CSF [172]. Therefore, M-CSF is an essential factor for the formation and activation of osteoclasts [3]. In a previous report, primary osteocyte and osteoclast precursors treated with 1,25(OH)_2_D_3_ and PGE2 showed a high level of RANKL and osteoclast formation [12]. These results suggest that 1,25(OH)_2_D_3_ and PGE2 also induce M-CSF in osteocytes. In our previous study, a coculture system of highly purified Dmp1-Topaz primary osteocytes obtained by cell sorting and TNF receptor I- and II-deficient osteoclast precursors treated with TNF-α showed a significant increase in TRAP-positive cells, whereas cultures without TNF-α did not show TRAP-positive cells. TNF-α directly affects osteocyte RANKL expression and increases osteoclastogenesis. However, TNF-α does not enhance M-CSF expression in osteocytes [16]. These results suggest that the constitutive level of M-CSF in osteocytes induces osteoclast formation.

### 3.9. VEGF

The VEGF family includes six homodimeric proteins: VEGF-A, VEGF-B, VEGF-C, VEGF-D, VEGF-E, and placenta growth factor. VEGF is the most important mediator regulating vascular development and angiogenesis [173]. It has been reported that osteoblasts and chondrocytes secrete VEGF in bone and that VEGF plays a critical role in skeleton development [174]. VEGF derived from hypertrophic chondrocytes binds to VEGFR1 in monocytes during bone development to promote monocyte migration and differentiation to osteoclasts [175]. It has been reported that VEGF binds to VEGFR2 in osteoclasts and promotes osteoclast activity through PI3K/Akt signaling stimulation [176]. It has also been reported that VEGF can replace M-CSF to promote osteoclast differentiation and bone resorption in vivo and in vitro [177]. Furthermore, VEGF enhances the bone resorptive activity of osteoclasts [178]. VEGF-A is activated by advanced glycation end products via ERK1/2, P38, and STAT3 in patients with diabetes and the elderly through upregulation of osteocyte-derived IL-6 and secretion [125], and osteocyte apoptosis [125,126]. Chronically elevated glucose levels and activator of G-protein signaling are linked to osteoporosis and a disruption in bone cell activity. Several factors from apoptotic osteocytes also induced nearby osteocytes and macrophages to release VEGF, RANKL, TNF-α, IL-6, and IL-1β [73].

### 3.10. Sclerostin

SOST and its gene product, Sclerostin, were identified by examining genes involved in sclerosteosis [179]. Sclerostin is a secreted glycoprotein that is primarily expressed in osteocytes and acts as a negative regulator of bone homeostasis through inhibition of bone formation by osteoblasts [180]. It has been reported that sclerostin is expressed in not only osteocytes, but also cementocytes, hypertrophic chondrocytes, osteoclasts, and periodontal ligament cells [181,182]. Sclerostin binds to LRP5/6 as an antagonist of canonical Wnt signaling, which inhibits bone formation [183]. The sclerostin antibody Evenity^®^ (Romosozumab) was recently approved for clinical use in the US for the treatment of postmenopausal women with osteoporosis, who have a high risk of fracture [184]. However, sclerostin antibody treatment may promote TNF-dependent inflammatory joint destruction in rheumatoid arthritis patients. Therefore, caution should be taken when using an anti-sclerostin antibody in patients with TNF-dependent rheumatism [185]. Sclerostin has been reported to increase RANKL expression in MLO-Y4 osteocytes. It has been suggested that sclerostin is capable of promoting osteoclast formation and osteoclast resorptive activity [186]. TNF-α upregulates the expression of expression in MLO-Y4 cells [98]. In our recent study, TNF-α enhanced the expression of sclerostin in highly purified primary osteocytes and sclerostin-induced RANKL expression in primary osteocytes enhanced osteoclast formation [17]. Consistent with these findings, the use of a TNF-α antagonist reduces RANKL and sclerostin expression in osteocytes of periodontitis rats with diabetes [76]. Sclerostin gene knockout also attenuates alveolar bone loss in mice with periodontitis [77]. Periodontitis-induced sclerostin elevates the RANKL/OPG ratio and ERK1/2 in alveolar bone [77]. Antagonizing TNF-α using infliximab reduces expression of sclerostin and RANKL in osteocytes and the alveolar bone loss in diabetic rats with periodontitis [76]. Inhibiting SOST expression using the unique design of zoledronic acid-loaded nanoparticles targeting osteocytes attenuates early breast cancer metastasis to bone [82]. Similar to other inflammatory cytokines, sclerostin expression is also upregulated in MLO-Y4 cells in the settings of high glucose and advanced glycation end products [126]. High glucose levels increase the production of reactive oxygen species, osteocyte apoptosis, and TNF-α that in turn upregulates sclerostin expression in osteocytes [101]. A high level of sclerostin in chronic kidney disease patients is associated with the pathogenesis of bone disorders such as an increased fracture risk and correlates positively with serum 1,25(OH)_2_D_3_, phosphorus, and TNF-α, indicating a decrease in the bone turnover rate [187,188]. Serum RA enhances SOST gene expression in human osteocyte-enriched cell cultures [28]. Bone loss and an increase in sclerostin expression by osteocytes have been reported in inflammatory bowel disease [31], spinal cord injury [86], and focal radiotherapy [189].

### 3.11. IL-10

IL-10 is pleiotropic cytokine that plays an important role in immunoregulation and inflammation. It is produced by B cells, mast cells, eosinophils, macrophages, and dendritic cells [190]. IL-10 is also produced from T helper 2 cells and downregulates the expression of cytokines in T helper 1 cells, such as IL-1, IL-6, and TNF-α [191]. Therefore, IL-10 is recognized as an anti-inflammatory cytokine. IL-10 inhibits osteoclast formation via a direct action on osteoclast precursors [54,192]. It inhibits proinflammatory cytokine production in inflammatory bone resorption sites [193]. Therefore, IL-10 suppresses osteoclast formation and bone resorption. Spinal cord injury induces bone loss by elevating the levels of circulating proinflammatory cytokines. In osteocytes, these cytokines include TNF-α, IL-6, IL-17, IL-10, RANKL, and sclerostin, which lead to osteoclast formation and bone resorption [86]. Because all of these cytokines except for IL-10 induce osteoclast formation and bone loss, it is still unclear which role IL-10 plays in spinal cord injury. These conflicting results require further experimentation. 

### 3.12. IL-17

The IL-17 family of cytokines is well acknowledged for its essential role in inflammation and immune system regulation. The IL-17 family includes six cytokines: IL-17A (IL-17), IL-17B, IL-17C, IL-17D, IL-17E (IL-25), and IL-17F [194]. IL-17A, commonly referred to as IL-17, is the most widely investigated member, owing to its proinflammatory properties. This cytokine is mainly produced by T helper 17 (Th17) cells in addition to others immune cell types such as γδT cells, neutrophils, innate lymphoid cells, macrophages, mast cells, natural killer cells, natural killer T cells, and B cells [195,196,197,198]. IL-17 activates multiple downstream signaling cascades, including mitogen-activated protein kinases (MAPKs) ERK1/2, JNK, P38, NF-κB, STAT3, and Nrf2/keap1 [199,200,201], which further induce upregulation of several inflammatory cytokines and chemokines such as TNF-α, IL-23, IL-1β, IL-6, IL-8, CCL4, CCL20, CXCL1, and CXCL12 [199,202,203,204,205]. Th17 cells have osteoclastogenic effects directly or indirectly through IL-17-mediated stimulation of osteoclast-related molecules on a wide variety of target cells. IL-17 treatment of CD14+ cells upregulates expression of osteoclastogenic genes, such as TRAP, c-fms, and RANK, which results in an increased number of TRAP+ multinucleated cells [206]. In addition, IL-17 directly promotes osteoclast formation and activation of CD11b+ cells in the absence of osteoblasts or exogenous sRANKL by increasing TNF-α and RANKL secretion from IL-7-treated monocytes [207]. It has also been reported that IL-17-induced RANKL mediates osteoclast formation by fibroblast-like synoviocytes isolated from adjuvant induced arthritis rats through the IL-17/IL-17RA/STAT-3 signaling cascade [208]. Furthermore, IL-17 enhances osteoblast-mediated bone resorption by promoting RANKL, M-CSF, and PGE2 activities, which in turn leads to increased cathepsin K and MMP-9 expression [209]. Osteocytes are believed to play the main role in regulating bone remodeling by sensing and responding to mechanical stimuli and have been demonstrated to promote osteoclastogenesis. Recent studies have shown that both IL-17 and IL-17RA are expressed in MLO-Y4 osteocytes. IL-17 increases the proliferation of MLO-Y4 cells and has no effect on MLO-Y4 cell apoptosis. MLO-Y4 cells incubated with IL-17 show increased levels of RANKL and TNF-α mRNA expression and sRANKL secretion. ERK1/2 and STAT3 signaling is inhibited during IL-17A-induced osteoclastogenesis, while the ephrinA2-EphA2 pathway is activated [210,211]. Moreover, new evidence has demonstrated that IL-17/IL17RA signaling in osteocytes is needed for parathyroid hormone to exert its regulatory effect on bone. IL-17 serves as an upstream cytokine that elevates the sensitivity of osteocytes to parathyroid hormone and enhances osteocytic RANKL secretion induced by parathyroid hormone. In addition, an in vivo study using IL-17RA knockout mice lacking expression of IL-17RA in DMP1-cre-expressing cells revealed that continuous parathyroid hormone supplementation does not induce cortical and trabecular bone loss and attenuates the capability of parathyroid hormone to promote osteocytic RANKL production [212]. Psoriasis is a chronic skin condition characterized by inflammation and is associated with high serum IL17A [213]. The Wnt signaling pathway in osteocytes and osteoblasts was blocked in a mouse model of psoriasis, which was mediated by the action of IL-17A, leading to arrest of bone formation in vivo rather than activation of osteoclast formation [32]. IL-17 is also expressed during spinal cord injury and leads to bone loss [86].

### 3.13. Parathyroid Hormone

PTH is a major regulator of extracellular calcium and phosphate homeostasis [214]. PTH has a dual effect on bone, both anabolic and catabolic. Osteocytes are regarded as a major target for PTH [215]. In cases of chronic PTH elevation, such as in hyperparathyroidism, there is an increase in bone turnover tipping towards a net loss of bone mass. Paradoxically, PTH is considered an anabolic drug in osteoporosis cases when administered only intermittently due to enhanced osteoblast differentiation and activity, as well as reduced osteoblast apoptosis [27]. In mice lacking PTHr under the control of the 10-kb Dmp1 promotor, there was a decrease in trabecular bone and osteopenia [216]. PTHr deletion in both the 8- and 10-kb Dmp1 promotor models resulted in decreased RANKL/OPG ratio, therefore, PTH signaling is important for maintaining RANKL relative levels that drive osteoclast formation [217]. PTH decreases SOST and sclerostin expression [218,219], providing another mechanism by which PTH control osteocyte bone remodeling. Activation of PTHr signaling in osteocytes leads to suppression of sclerostin expression, and activation of Wnt pathway signaling by osteocytes in vivo [215]. Mice lacking the PTHr receptor expression in osteocytes in a Dmp1 10 kb-Cre mice show increased sclerostin levels [216]. PTHr signaling controls both bone anabolism and catabolism via inhibition of sclerostin and activation of RANKL expression, respectively.

## 4. Conclusions

Osteocytes are the most abundant cell type in bone and outlive other bone cell populations. They communicate with each other via an extensive network of dendrites, resembling an organ, which allows osteocytes to moderate the inflammatory process locally and systemically (Figure 1). Osteocytes act by secreting molecules that influence the osteocyte itself, the immediate vicinity of the osteocyte, or systemically, behaving as a true endocrine cell. Osteocytes respond to a wide array of effector cytokines and in turn produce more cytokines to augment or halt inflammation and bone resorption. Osteocytes were once thought to be dormant, but are now in the spotlight and should be considered as a target cell for therapies of bone diseases. Osteocytes perform functions that were thought to be reserved for other specialized cell types, and future studies may reveal much more information on the osteocyte-bone dynamic and possibly confirm osteocytes as target cells for drugs of bone resorption and other systemic inflammatory diseases.

## Figures and Tables

**Figure 1 ijms-21-05169-f001:**
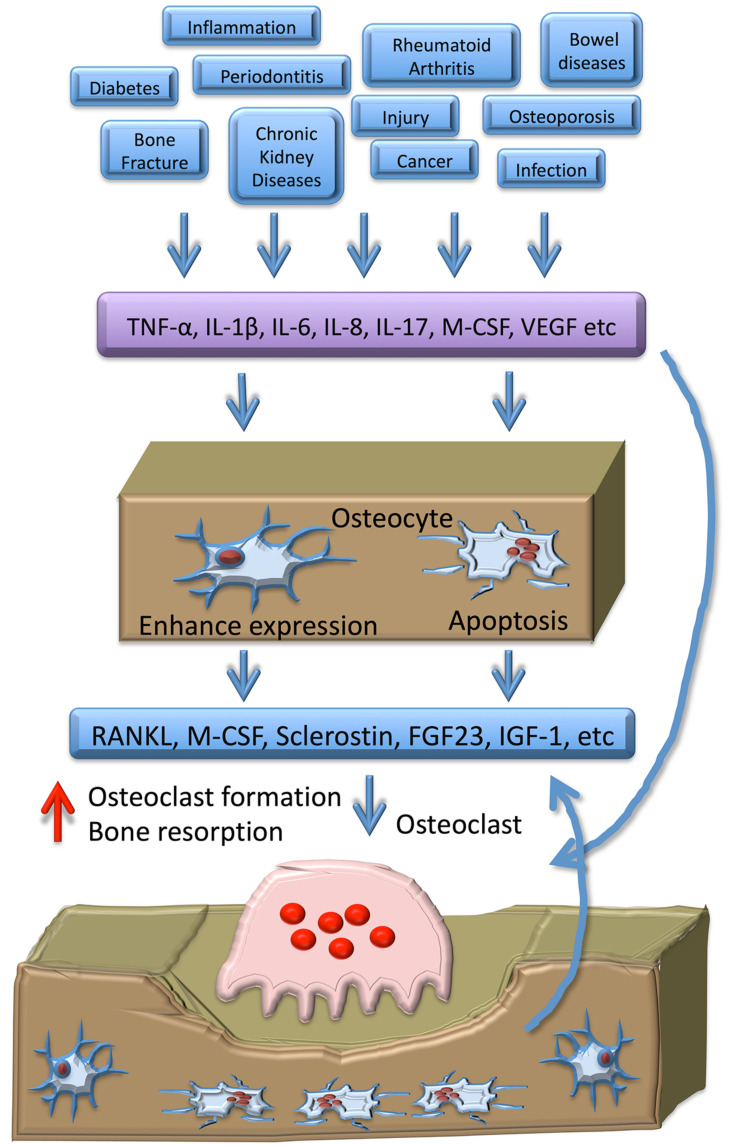
Schema of osteocyte-related cytokines that regulate osteoclast formation and bone resorption. Several cytokines are induced under pathological conditions. These cytokines directly induce osteoclast formation and bone resorption, as well as induce expression of osteoclastogenic cytokines in osteocytes and apoptosis of osteocytes as a result of released osteoclastogenic cytokines. These cytokines induce osteoclast formation and bone resorption. In the process of bone resorption, apoptosis of osteocytes is induced, which releases osteoclastogenic cytokines.

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
