# Peer review of "Osteocyte-Related Cytokines Regulate Osteoclast Formation and Bone Resorption"

_ijms, 2020, doi:10.3390/ijms21145169_

Round 1

Reviewer 1 Report

In this review Kitaura et al. provide an overview of osteocyte-related cytokines and hormones that influence the activity of bone resorbing osteoclasts. The manuscript is very well written and researched and indeed quite comprehensive. I believe this could be a valuable resource for researchers wishing to learn more about the cytokine-related regulation of osteoclasts via osteocytes. I recommend the publication of this manuscript, given that the points listed below are addressed:

In the introduction rheumatoid arthritis and periodontitis are given as examples for disorders due to imbalanced formation/resorption. Given its extraordinarily high incidence, osteoporosis should be mentioned as well.

The authors should briefly introduce the concept of osteocytic osteolysis. Likewise, the statement that osteoclasts are the only cells capable of bone resorption is misleading and should be rephrased.

In paragraph 3.1 (RANKL and OPG) the authors should briefly discuss the dual nature of Rankl as a membrane bound protein that can be released under specific circumstances. This would help the reader with interpreting results that are based on altered expression (but not necessarily release) of Rankl.

Cx43 is a very well established gap junction channel and allows the passage of many signaling molecules. However, its function as a signaling protein itself is less clear. Thus, line 139 could be more carefully phrased as: “These results suggest that signaling through Cx43 controls osteocyte survival.”

The later part of paragraph 3.1 contains a listing of many disorders influencing osteocyte Rankl levels. Here also the effect of burn injuries should be included.

In paragraph 3.5 it should be mentioned that the mechanism by which altered Phex activity influences Fgf23 levels is not fully elucidated, since Fgf23 is not a substrate of Phex (Liu et al., 2003, JBC). Indeed, there is some data indicating that the mechanism is mediated via osteopontin (Barros et al., 2013, JBMR).

The authors should include a paragraph about the effect of PTH, as it is one of the few Rankl-inducing messenger molecules that actually also increases the release of Rankl from the cell surface. Indeed, many cytokines, such as Tnf-a, IL-6 and IL-17 do indeed increase the expression, but not necessarily the shedding of Rankl, at least in osteoblasts (Heckt et al., 2016, Bone).

Generally, while clearly not a focus of this manuscript, the authors could provide some information on the osteoanabolic functions of osteocytes (maybe in the introduction), providing a more balanced overview.

The antibody that is the active ingredient in the medication Evenity is spelled “Romosozumab”. Likewise, please check, if Evenity needs to be shown with a trademark symbol (Evenity®).

Figure 1: The white font is hard to read on the light blue/ light violet background.

If possible, please show some interconnected osteocytes in the figure.

Author Response

Thank you very much for your valuable opinion and we sincerely appreciate your valuable feedback. All the questions, we thought them over carefully. These are our responses as follows.

  1. In the introduction rheumatoid arthritis and periodontitis are given as examples for disorders due to imbalanced formation/resorption. Given its extraordinarily high incidence, osteoporosis should be mentioned as well.

Response: Thank you for your comment, osteoporosis is added to the text. Lines 19, 41.

  1. The authors should briefly introduce the concept of osteocytic osteolysis. Likewise, the statement that osteoclasts are the only cells capable of bone resorption is misleading and should be rephrased.

Response: A paragraph explaining the concept osteocytic osteolysis is added. Line 73. Osteoclasts as the only cell capable of bone resorption is modified to: Osteoclast is he principal cell of bone resorption. Lines 18 and 39.

  1. In paragraph 3.1 (RANKL and OPG) the authors should briefly discuss the dual nature of Rankl as a membrane bound protein that can be released under specific circumstances. This would help the reader with interpreting results that are based on altered expression (but not necessarily release) of Rankl.

Response: A paragraph about the dual nature of RANKL as a soluble and membrane-bound forms is added. Line 196.

  1. Cx43 is a very well established gap junction channel and allows the passage of many signaling molecules. However, its function as a signaling protein itself is less clear. Thus, line 139 could be more carefully phrased as: “These results suggest that signaling through Cx43 controls osteocyte survival.”

Response: The text is changed to channeling through Cx43. Line 171.

  1. The later part of paragraph 3.1 contains a listing of many disorders influencing osteocyte Rankl levels. Here also the effect of burn injuries should be included.

Response: The effect of burn injuries is added. Line 190.

  1. In paragraph 3.5 it should be mentioned that the mechanism by which altered Phex activity influences Fgf23 levels is not fully elucidated, since Fgf23 is not a substrate of Phex (Liu et al., 2003, JBC). Indeed, there is some data indicating that the mechanism is mediated via osteopontin (Barros et al., 2013, JBMR).

Response: The indirect nature of he relationship between PHEX and FGF 23 has be explained in more detail, with possible scenarios of how PHEX could alter FGF 23 levels. Lines 328, 334.

  1. The authors should include a paragraph about the effect of PTH, as it is one of the few Rankl-inducing messenger molecules that actually also increases the release of Rankl from the cell surface. Indeed, many cytokines, such as Tnf-a, IL-6 and IL-17 do indeed increase the expression, but not necessarily the shedding of Rankl, at least in osteoblasts (Heckt et al., 2016, Bone).

Response: PTH regulation of RAKL an sclerostin expression by osteocytes is explained in line 547.

  1. Generally, while clearly not a focus of this manuscript, the authors could provide some information on the osteoanabolic functions of osteocytes (maybe in the introduction), providing a more balanced overview.

Response: While the concept of osteocyte osteoanabolic potential is relatively rare, we provided information on how osteocytes through expression of sclerostin can alter bone formation status. Line 83.

  1. The antibody that is the active ingredient in the medication Evenity is spelled “Romosozumab”. Likewise, please check, if Evenity needs to be shown with a trademark symbol (Evenity®).

Response: The name is corrected to Romosozumab. And the ® trademark is added to become Evenity®.

  1. Figure 1: The white font is hard to read on the light blue/light violet background.

Response: We changed white font to black font and some parts clearly.

  1. If possible, please show some interconnected osteocytes in the figure.

Response: We made graphic abstract including interconnected osteocytes.

Reviewer 2 Report

This review article by Kitaura and colleagues discusses recent work into the many cytokines that influence osteoclastogenesis and bone resorptive activities. It represents a very comprehensive treatment of this important topic. The paper is logically organized and clearly written.

Author Response

Thank you very much for your valuable opinion and we sincerely appreciate your valuable feedback. All the questions, we thought them over carefully. These are our responses as follows.

  1. This review article by Kitaura and colleagues discusses recent work into the many cytokines that influence osteoclastogenesis and bone resorptive activities. It represents a very comprehensive treatment of this important topic. The paper is logically organized and clearly written.

Response: Thank you very much for your comment.

Reviewer 3 Report

In this review, the authors describe the role of osteocytes in bone remodeling in an exhaustive and detailed manner.

The response of osteocytes to a wide range array is explained in detail and the production of cytokines and their fundamental role in increasing or stopping inflammation and bone resorption is also perfectly explained. Hence, it is clear, how osteocytes could be considered as a target cell for different bone disease therapies.
The authors could add some schemes, to the one already present, to describe and immediately make clear the effect of the cytokines and their possible main importance, compared to each other

Author Response

Thank you very much for your valuable opinion and we sincerely appreciate your valuable feedback. All the questions, we thought them over carefully. These are our responses as follows.

  1. The authors could add some schemes, to the one already present, to describe and immediately make clear the effect of the cytokines and their possible main importance, compared to each other.

Response: Thank you very much for your comment. We also think so. In this review, we would like to just show which cytokines from osteocyte concerned with osteoclast formation and which cytokines affect cytokines expression in osteocyte. Next time, we will show comparing to each other.